



# Historical porosity data in polar firn

Kévin Fourteau[1], Laurent Arnaud[1], Xavier Faïn[1], Patricia Martinerie[1], David M. Etheridge[2], Vladimir Lipenkov[3], and Jean-Marc Barnola[†1]

[1]Univ. Grenoble Alpes, CNRS, IRD, Grenoble INP, IGE, F-38000 Grenoble, France
[2]Climate Science Centre, CSIRO Oceans and Atmosphere, Aspendale, Victoria, Australia
[3]Climate and Environmental Research Laboratory, Arctic and Antarctic Research Institute, St. Petersburg, 199397, Russia
[†] deceased, September 21, 2009

**Correspondence:** laurent.arnaud@univ-grenoble-alpes.fr

**Abstract.** In the 1990's, closed and open porosity volumes of firn samples have been measured by J.-M. Barnola using the technique of gas pycnometry, on firn from three different polar sites. They are the basis of a parameterization of closed porosity in polar firn, first introduced in Goujon et al. (2003) and used in several firn physics models (e.g. Buizert et al., 2012). However, these data and their processing have not been published in their own right yet. In this short article, we detail how

they were processed by J.-M. Barnola, and how the closed porosity parameterization was obtained. We show that the original data processing only partially accounts for the presence of re-opened bubbles in the samples. Since the proper correction to apply for this effect is hard to estimate, we also processed the data without including a correction for re-opened bubbles. Finally, we intend to make these pycnometry data available, in order to be used by the glaciology community, notably for the study of polar ice formation and of the composition of gas records in ice cores. They are hosted on the PANGAEA database:

https://doi.pangaea.de/10.1594/PANGAEA.907678 (Fourteau et al., 2019a).

## 1 Introduction

The enclosing of atmospheric air in the ice of polar regions is of great importance for the study of past climates. Indeed, ice cores drilled in the polar regions have the unique characteristics of containing bubbles of air from past atmospheres. They have

15 thus been used to reconstruct the atmosphere history in major greenhouse gas concentrations for the last $800,000$ years (Lüthi et al., 2008; Loulergue et al., 2008). However, in order to properly interpret the gas records from ice cores, it is necessary to understand the trapping of air in the ice (Schwander and Stauffer, 1984; Stauffer et al., 1985; Schwander et al., 1993; Rommelaere et al., 1997; Trudinger et al., 2002).

The snow at the surface of polar ice sheets is a porous material, and interstitial air can freely exchange with the atmosphere

(Stauffer et al., 1985). Snow strata are progressively buried under new precipitations and are compressed due to the weight of the younger snow above. This buried, metamorphosed, and compacted snow is referred to as firn. With time the firn strata are



**Table 1.** Characteristics of the three investigated polar sites

| Site Name | Location | Accumulation($\mathrm{g\,cm^{-3}.yr^{-1}}$) | Temperature ($^\circ$C) |
|---|---|---|---|
| Vostok | East Antarctic plateau | 2.2[a] | -56[a] |
| Summit | Central Greenland | 20.9[b] | -31[b] |
| DE08-2 | Coastal East Antarctica | 110[c] | -19[d] |

a: Lipenkov et al. (1997)
b: Schwander et al. (1993)
c: Etheridge et al. (1996)
d: Etheridge et al. (1992)

further buried and their interstitial porous networks shrink. Depending on the local temperature and accumulation conditions, some of the pores start to pinch and encapsulate the interstitial air at depths between 50 and 100m below the surface (Witrant et al., 2012). The porous network then continues to close until all the interstitial air is isolated from the atmosphere. The firn then becomes airtight ice with enclosed bubbles of atmospheric air.

One way to characterize the closing of the porous network and the trapping of gases is to measure the closed and the open pore volumes. Closed pores are pores that no longer reach to the atmosphere, and are therefore airtight. On the other hand, open pores reach up to the atmosphere, through an interconnected porous network. Moreover, data of closed and open pore volumes are required for the usage of gas trapping models (Rommelaere et al., 1997; Goujon et al., 2003; Buizert et al., 2012; Witrant et al., 2012). That is why closed and open volumes have been measured along various firn columns from Greenland

and Antarctica (Schwander and Stauffer, 1984; Schwander et al., 1993; Schaller et al., 2017).

Such measurements were notably performed in the 1990's by J.-M. Barnola on firn cores drilled at the three polar sites of Vostok (Antarctica), Summit (Greenland) and DE08-2 (Antarctica), using the technique of gas pycnometry (Schwander and Stauffer, 1984; Stauffer et al., 1985). The three sites have very different characteristics, from the cold and low-accumulation site of Vostok to the high-accumulation site of Law Dome's DE08-2. Their respective accumulation rates and temperatures

are given in Table 1. The three sites have been exploited for the range of environmental information they contain, from recent decades to more than 400,000 years before present (Barnola et al., 1987; Schwander et al., 1993; Trudinger et al., 2002). The obtained porosity data have been widely used to parameterize closed porosity as a function of density in firn physics models (Trudinger et al., 2002; Goujon et al., 2003; Buizert et al., 2012; Witrant et al., 2012). Unfortunately, J.-M. Barnola passed away before publishing the data in the peer-reviewed literature. For transparency, and to recognize J.-M. Barnola's effort, care

and foresight in undertaking the measurements at three remote sites, we decided to make them available. Our goal is also to provide an explanation of how the closed porosity parameterization proposed in Goujon et al. (2003) was derived. Moreover, measuring closed porosity is labor intensive and requires a large amount of firn material. As a result, this type of data is rather scarce. We hope that making these data available will help the ice core community to better understand the trapping of gases in polar ice.





## 2 The pycnometry method

The technique used by J.-M. Barnola to measure the closed and open porosity volumes in firn samples is the gas pycnometry method (Schwander and Stauffer, 1984; Stauffer et al., 1985). The pycnometry apparatus is composed of two airtight chambers of known volumes $V_1$ and $V_2$ with a valve between them allowing to either connect or isolate the chambers. A scheme is

provided in the Supplementary Material of Fourteau et al. (2019b). A pressure gauge is joined to the chamber $V_1$ to monitor its internal pressure. For the measurements, a firn sample is placed in the first chamber $V_1$, while the second one is isolated and vacuum-pumped. Placing a firn sample in the chamber $V_1$ renders a volume $V_s$ inaccessible to the gases. The pressure $P$ in $V_1$ is recorded. Then, the two chambers are connected, allowing the gas in chamber $V_1$ to expand in a larger volume. The pressure after expansion $P'$ is then recorded. The system was designed to minimize the pressure drop, in order to avoid

rupturing recently closed and still fragile pores. The volume $V_s$ can be related to the recorded pressure by:

$$V_s = V_1 - \frac{R}{1-R}V_2 \tag{1}$$

where $R = P'/P$.

The protocol followed by J.-M. Barnola was to first execute an expansion without any sample in the first chamber, then a second expansion with the sample in. In this case, Equation 1 can be rearranged as:

$$V_s = V_2\Big(\frac{R_0}{1-R_0} - \frac{R_1}{1-R_1}\Big) \tag{2}$$

where $R_0$ and $R_1$ are respectively the pressure ratio in the cases without and with the firn sample in $V_1$.

The volume inaccessible to gases is composed of the ice phase and of the closed pores phase. Therefore, one can deduce the closed and open porosity volumes:

$$
\begin{aligned}
V_{cl} &= V_s - V_{ice} \\
V_o &= V_{cyl} - V_s
\end{aligned} \tag{3}
$$

where $V_{cl}$ and $V_o$ are the closed and open porosity volumes, $V_{ice}$ the volume of the ice phase and $V_{cyl}$ the volume of the firn sample. The volume of the ice phase is deduced from the mass $M$ of the sample knowing that $V_{ice} = M/\rho_{ice}$, where $\rho_{ice}$ is the density of pure ice. The density of pure ice is estimated using the temperature relationship $\rho_{ice} = 0.9165(1 - 1.53 \times 10^{-4}T)$, where $\rho_{ice}$ is expressed in $\mathrm{g\,cm^{-3}}$ and $T$ is the temperature expressed in $^\circ C$ (Bader, 1964; Goujon et al., 2003). The volume of the firn sample $V_{cyl}$ is measured geometrically with calipers.

Note that in the pycnometry experiment, all the pores reaching the edge of the sample are considered as open. This means that some pores that are closed in the firn column (they do not reach the atmosphere) will be considered open during the pycnometry measurement. This is known as the cut-bubble effect and leads to an underestimation of the closed porosity (Martinerie et al., 1990; Schaller et al., 2017).





The firn samples used for the pycnometry measurements are cylindrical samples of about 4 to 5cm in height and diameter. They were produced by machining on a lathe and trimmed with a drop saw in order to produce well-shaped cylinders. The measurements were performed in environments with a good temperature stability, to limit the effect of temperature variations. Finally, in order to avoid post-coring effects, all the samples were measured directly on the field, shortly after the drilling of

the firn core.

## 3  Processing of the data

For each of the three sites we retrieved a computer file containing the expansion ratios $R_0$ and $R_1$, the mass of the firn samples, their volumes, and the temperature during the experiment. We also retrieved the source codes that J.-M. Barnola used to process the data. Finally, we have the experiment notebooks of J.-M. Barnola.

### 3.1  Original data processing

In this section, we aim to reproduce the data processing performed by J.-M. Barnola. This is done for two reasons. First, the data have been used to derive a parameterization of the closed porosity in polar firn (Goujon et al., 2003), and it is therefore important to understand how they were processed. Second, the original processing includes corrections for experimental biases that were observed by J.-M. Barnola and that have to be taken into account.

The original source codes indicate that the processing included a correction to account both for a pycnometry system drift and the cut-bubble effect. This correction is based on the idea that the pycnometry method should ideally indicate fully open samples at low density ($\rho < 0.72$g.cm$^{-3}$) and fully closed ones at high density ($\rho > 0.86$g.cm$^{-3}$). A correcting factor $\alpha$ to be applied to the inaccessible volume $V_s$ can thus determined for each of these low and high-density samples. The $\alpha$ factors

are such that the measurements of low-density firn samples (respectively high-density firn samples) yields a fully open (respectively closed) porosity. In the case of low-density firn, $\alpha$ is computed as $1 - V_{\mathrm{cl}}/V_{\mathrm{ice}}$, and in the case of high density firn $\alpha = V_{\mathrm{cyl}}/V_{\mathrm{s}}$. J.-M. Barnola observed that the $\alpha$ factors are linearly related to the empty expansion ratio $R_0$ (Figure 1). This might come as a surprise as $R_0$ is measured before the sample is inserted in the apparatus. Our understanding is that it reflects that $\alpha$ and $R_0$ are simultaneously affected by system drifts. By monitoring the evolution of $R_0$, one is able to estimate the

correcting factor $\alpha$ to be applied. However, as seen in Figure 1, the linear relationship is not the same for high density and low density firn samples. J.-M. Barnola derived intermediate linear relationships, in black in Figure 1. Thus, for each firn sample (including mid-density samples), a correcting factor is determined thanks to the preceding empty expansion and applied to the inaccessible volume $V_s$. Finally, in all of the three measurement campaigns the volume $V_2$ has been estimated to be of 7.2cm$^3$ (value found directly in J.-M. Barnola's processing codes).

Vostok measurement campaign:



The Vostok measurements were performed on the BH3 firn core, drilled during the austral summer 1991/92. For this measurement campaign, J.-M. Barnola observed a bias due to the measurement of sample volumes with a caliper, depending of the pressure applied by the caliper. He therefore proposed to apply a volume correction of $0.9\%$ to the firn samples that were measured by applying a too weak pressure with the caliper. He also applied a $4°C$ correction to the recorded temperatures, in order to account for the heat dissipation of heated elements towards the temperature sensor. For the Vostok campaign measurements $\alpha$ is computed following $\alpha = -40.2145R_0 + 39.22304$, shown in black in the left panel of Figure 1. These three corrections were found hard coded in the Vostok processing source code, and have been corroborated by the notebooks.

Summit measurement campaign:

The firn porosity measurement of Summit were performed in the framework of the 1989 Eurocore project. For this campaign the same volume correction of $0.9\%$ was applied to all the samples. Moreover, a weighting bias was found by J.-M. Barnola and is taken into account by applying a $0.9983$ correction factor to the measured mass. Finally, the $\alpha$ correction is given by $\alpha = -45.045045R_0 + 43.820459$, in black in the middle panel of Figure 1. It is interesting to note that for Summit the correction chosen by J.-M. Barnola seems to be primarily based on the low density $\alpha$ only.

DE08-2 measurement campaign:

The DE08-2 measurements were performed during the austral summer 1992/93. For DE08-2, no volume or mass correction is reported in the original processing code. The $\alpha$ correction is given by $\alpha = -37.2577R_0 + 36.4204$, in black in the right panel of Figure 1. Again, the correction chosen by J.-M. Barnola appears to be primarily based on the low density $\alpha$.

From these data, we can deduce the closed and open porosity volumes. Figure 2 displays closed porosity and closed porosity ratio values against total porosity. Closed porosity is defined as the volume fraction occupied by the closed pores in the firn sample, total porosity is defined as the volume fraction of all pores and the closed porosity ratio is the ratio of the closed pores volume over the total porous volume. We chose to use volume fractions instead of porous volumes per gram of firn, as the former are not sensitive to temperature and therefore renders the comparison between sites easier. Yet, the volume fraction data can easily be converted to porous volumes per gram of firn using the density of pure ice.

## 3.2 Uncertainty analysis

Unfortunately, the original data we retrieved do not allow us to perform a systematic uncertainty analysis. Indeed, we did not retrieve a quantification of the uncertainties of the raw measurements, such as the sample's mass or volume. We are therefore not able to propagate the uncertainties of the raw measurements to the final derived quantities, such as the density or the closed porosity ratio.

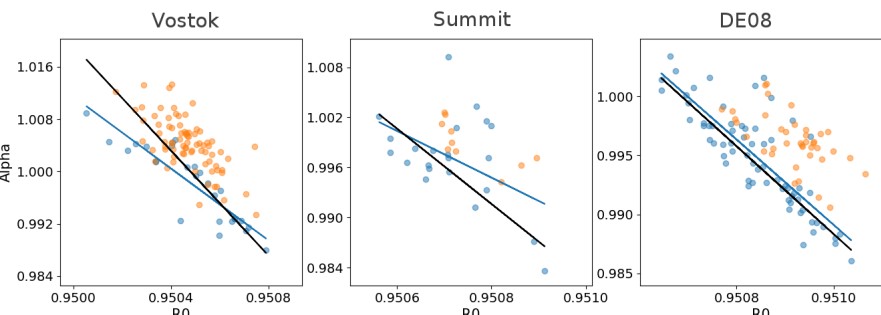

**Figure 1.** In each panel: the blue dots are the correcting factors $\alpha$ computed for low density firn as a function of the preceding $R_0$, the orange dots are the correcting factors $\alpha$ computed for high density firn as a function of $R_0$, the black line is the linear relation originally derived by J.-M. Barnola, and the blue line is the linear regression based solely on the low density $\alpha$

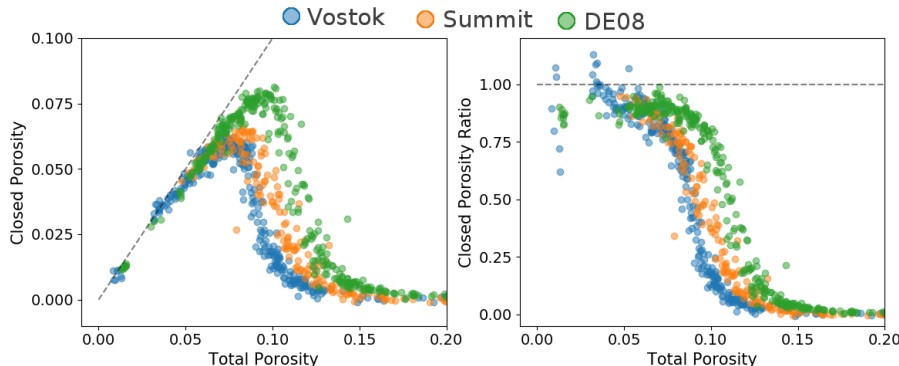

**Figure 2.** Porosity data obtained using J.-M. Barnola processing chain. Left panel: closed porosity against total porosity. The dashed line represents where the closed porosity equals the total porosity. Right panel: closed porosity ratio over total porosity. The dashed line indicates where the closed porosity ratio equals one.

However, a recent pycnometry campaign was conducted for an East Antarctic firn core, including a dedicated uncertainty analysis (Section S1.3 of the Supplement in Fourteau et al., 2019b). The measurements were performed with the same pycnometry apparatus and with the same sample size as the data presented in this article. We can therefore expect the data of JM Barnola to be affected by similar uncertainties. Fourteau et al. (2019b) quantified the errors associated with the measurements performed

5 in the cold room and used to derived the density and the closed porosity of the samples. This includes the errors on the mass of the sample, its radius and height, the pressures of the chambers, and the volumes of the chambers. These errors were then propagated to obtain an estimation of the uncertainty of the density and closed porosity of the samples.

Fourteau et al. (2019b) analysis indicates that the uncertainty on density is fairly constant over the entire range of measurements with a value of $0.0082\,\mathrm{g\,cm^{-3}}$, that is to say an uncertainty of about $0.009$ on density relative to pure ice. This represents a

10 relative uncertainty of about $1\%$ on the derived density. Note that this is of the same order as the correction applied by J.-M.

Barnola to the volumes of the Vostok samples. Contrary to density, the uncertainty of the closed porosity ratio is not constant over the entire range of data, and increases from about $0.02$ for low-density samples to about $0.2$ for high-density samples. For both quantities, the dominant contribution to the final uncertainty is the uncertainty of the measured sample volume.

### 3.3 A new data processing

We identify one major issue in the processing elaborated by J.-M. Barnola. In the case of high density firn samples, determining a correcting factor with $\alpha = V_{\mathrm{cyl}}/V_{\mathrm{s}}$ both encapsulates the effect of a system drift and of cut-bubbles. Indeed, the assumption under which this factor is computed is that the pycnometry experiment should measure a fully closed sample at high density, de facto including a cut-bubble correction. However, the correction to be applied for cut-bubbles is not the same at all densities (Schaller et al., 2017). It thus explains why the high and low density $\alpha$ relationship with $R_0$ might differ. On the other hand,

the low density $\alpha$ does not include any cut-bubble correction, and therefore should only account for system drifts. We therefore propose to correct the data using a linear regression between $R_0$ and the low density $\alpha$ only. These corrections are displayed as blue lines in Figure 1. As shown in the figure, the new corrections mainly differ in the Vostok case. The closed porosity and closed porosity ratio after applying this new correction are displayed in Figure 3.

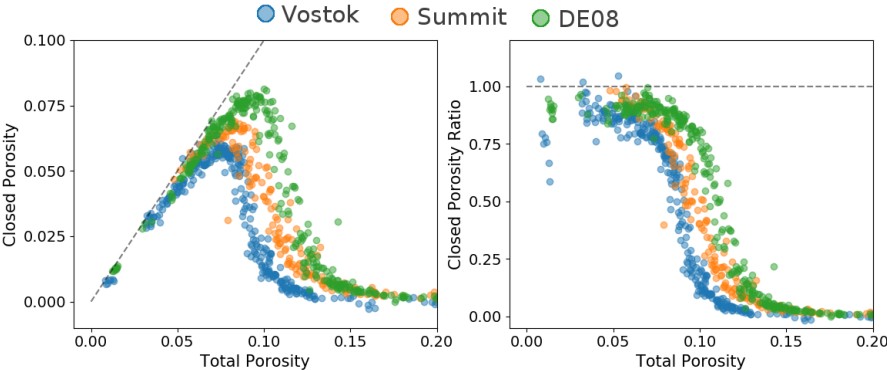

**Figure 3.** Same as Figure 2, with the new $\alpha$ correction.

It is important to note that these data are not corrected for cut bubbles, and therefore do not directly represent the amount

of closed pores in the firn column. We decided not to correct the data for cut bubbles in this article. Indeed, the appropriate corrections are hard to estimate and potentially site dependent (Schaller et al., 2017). Recently, Schaller et al. (2017) reported a fraction of re-open bubbles of up to $60\%$ for similar size B53 firn samples (East Antarctic plateau). Further research is needed to fully solve this problem.



## 4 The Barnola parameterization for closed porosity

The firn densification and gas trapping model of Goujon et al. (2003) uses a parameterization of closed porosity proposed by J.-M. Barnola (Equation 9 of Goujon et al., 2003). This Barnola parameterization relates the closed porosity to the total porosity with:

$$P_{\text{closed}} = \gamma P_{\text{total}} \left( \frac{P_{\text{total}}}{P_{\text{close-off}}} \right)^{-7.6} \tag{4}$$

where $P_{\text{closed}}$ is the closed porosity, $P_{\text{total}}$ the total porosity, $P_{\text{close-off}}$ the close-off porosity that can be estimated using air content measurements or a temperature regression (Martinerie et al., 1994), and $\gamma$ a factor valued at $0.37$.

We are confident that the Barnola closed porosity parameterization was deduced from the pycnometry data described in Section 3.1, with the original processing chain. Indeed, there is a clear linear relationship between the logarithm of the closed porosity ratio and the logarithm of the total porosity normalized by the porosity at mean close-off deduced from air content data. This is relation, displayed in Figure 4, is consistent with the Barnola parameterization. The comparison between the experimental closed porosities and the Barnola parameterization is also displayed in Figure 5. It is therefore important to acknowledge that the Barnola parameterization is based on data that are not fully corrected for cut-bubbles. Future users of this parameterization should be aware of this potential limitation. However, since we are not able to properly estimate the corrections to be applied for cut bubbles, we cannot propose a new law replacing the Barnola parameterization at this point.

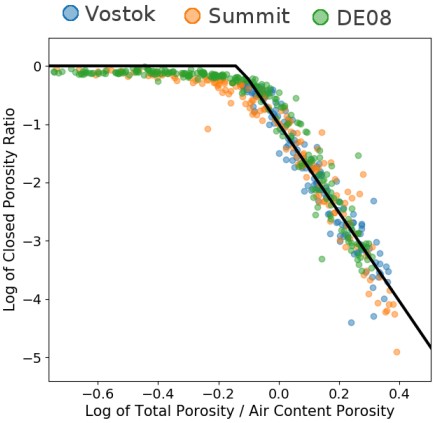

**Figure 4.** Relationship between the logarithm of the closed porosity ratio and the logarithm of the total porosity. The solid black line corresponds to the Barnola parameterization. The closed porosity data were obtained with the original processing chain.



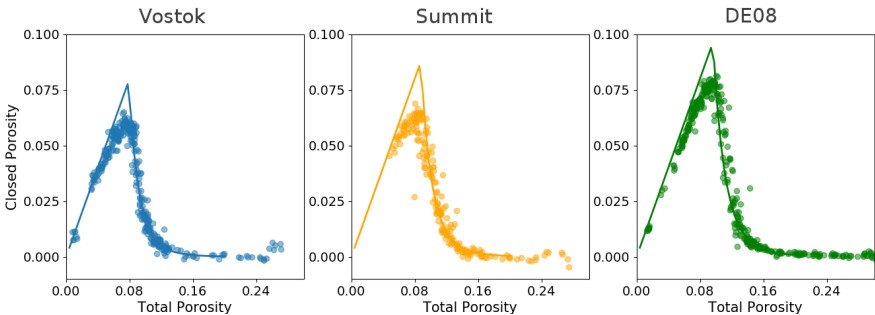

**Figure 5.** Measured closed porosity in the Vostok, Summit and DE08-2 firns with the original processing chain. The corresponding Barnola parameterizations are displayed as solid lines.

## 5   Conclusions

We evaluated the pycnometry data from three polar sites obtained in the 1990's by J.-M. Barnola. Based on original computer files, including raw data and processing source codes, we were able to reproduce the processing chain developed by J.-M. Barnola, including experimental bias corrections. We found that these data have not been fully corrected for the cut-bubble effect. We also confirm that the closed porosity data deduced from those pycnometry experiments were used to derive the Barnola closed porosity parameterization, first introduced in Goujon et al. (2003). Consequently, this parameterization suffers from the incomplete cut-bubble correction of the pycnometry data. More work is needed to quantify the amount of re-open bubbles in firn samples, but recent work highlighted a fraction of re-opened bubbles reaching up to $60\%$ (Schaller et al., 2017).

Finally, we intend to make these data publicly available (Fourteau et al., 2019a). The three sites studied in this article are characterized by a wide range of accumulation rates and temperatures. Such type of data are crucial to understand the age, amount and composition of the air enclosed in polar ice sheets. They could be useful for future studies focusing on the effect of the climatic conditions on pore closure and gas trapping, as well as to interpret long term ice core atmospheric records.

*Code availability.* The codes used to process the data were developped using python3. They will be provided upon direct request to the corresponding authors.

*Data availability.* The pycnometry datasets generated with the new correction methodology are hosted on the PANGAEA database: https://doi.pangaea.de/10.1594/PANGAEA.907678 (Fourteau et al., 2019a). The datasets generated using the original JM Barnola correction methodology will be provided upon direct request to the corresponding authors.



*Author contributions.*  The pycnometry measurements were performed by JMB with the help of DME and VL. The original data files and notebooks were retrieved by LA, XF and PM. The reconstruction of the original processing chain was done by LA, XF, KF and PM. The codes to process the data were developped by KF. All authors contributed to the interpretation of the data. The manuscript was written by KF with the help of all the co-authors.

5  *Competing interests.*  The author declare having no competing interests.

*Acknowledgements.*  We are grateful to Jakob Schwander for his help for the pycnometry measurements of the Summit site. We thank the 1991/92 Vostok, 1989 Eurocore, and 1992/93 DE08-2 at Law Dome ice core programs, as well as the field personal that contributed to these successful shallow drilling operations. We finally acknowledge Olivier Magand for his help retrieving the original notebooks, and his tests
10  of the pycnometry method.



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
