# Peer review of "Historical porosity data in polar firn"

_Earth System Science Data, 2019_

## Referee Comment (RC1) · Christo Buizert (Referee) · 17 Feb 2020

Fourteau et al. publish detailed historical firn pycnometry records from three sites (Vostok, Greenland Summit and Law Dome DE-08) that were never published due to the passing of Jean-Marc Barnola. A porosity parameterization based on these data (published in Goujon et al. 2003) is widely used, and the paper describes the underlying data. These are very valuable observations that need to be published, and I commend Fourteau et al. for taking the effort to do so. I only have some recommendations for technical corrections.

I am very intrigued to see the differences in the close-off density at these three sites (Fig. 3). This suggests highest air content at DE08, and lowest at Vostok. This is indeed the observation of Martinerie et al. (1994). Do you think this is the underlying mechanism of the Martinerie et al. relationship between air content and temperature?

[Figure]

This is certainly what is implied by Fig. 5. I think it would be appropriate to address this briefly in the paper.

Would it be possible to show the newer pycnometry data from the "Lock-In" site (Fourteau et al. 2019) somewhere together with the data from the three old sites (for example as a third panel in Fig. 3)? One would expect them to look more like Vostok than like Summit or DE08. Is that indeed the case? Showing these data would also give us a way to compare the consistency of the older and newer data.

Could you give some examples of mechanisms that can cause system drift in the pynometry measurements? Are you talking about vapor freezing onto the surfaces, snow getting stuck, etc? Or is it something else?

P1L8: remove "intend to"

P1L20: remove plural "s" from "precipitations"

P2L6 and P2L7: replace "reach" with "connect"

P2L6: airtight is not the right word, given that fugitive gases like Ne still escape. I think you can just remove the phrase "and are therefore airtight".

P4L4: "on the field" should be "in the field"

P5L2: "depending of" should be "depending on"

---

## Referee Comment (RC2) · Johannes Freitag (Referee) · 25 Feb 2020

Fourteau et al present historical porosity data of firn from three polar sites originally measured by J.-M. Barnola with the means of gas pycnometry several years ago. These data were the base of the important closed porosity-density parametrization widely used in firn gas models. Fourteau et al. provide the raw data and give insight into the original data processing. They confirm the use of the dataset for the closed-porosity parametrization first introduced in Goujon et al. (2003) and highlight its limitation. I appreciate the reworking of this fundamental data set.

The manuscript is well written with an appropriate introduction and description of the method. The data processing is described in required detail to enable the reader to follow the analysis. The data set is accessible via the given identifier and complete.

I highly recommend the manuscript for publication. In my opinion the data set will

encourage further methodological improvements and investigations on the cut-bubble-effect. Maybe it would be worthwhile to set up a future study where different methods like X-ray tomography and pycnometry are applied on the same set of samples?

Specific comments: 1) The names of the columns in the data set are a little bit misleading. The pore volumes are named by "Poros_xx" like "Poros_closed_cm__3_" which might be interpreted as abbreviation for porosity (no units) instead of pore volumes (unit in cm_3). I would suggest names like "Pore_vol_xx". 2) The column "Pores_frac" should be renamed to "ClosedPorosRatio" as it is defined in the manuscript. 3) In the data sets there are some non-physical values like negative pore volumina or closed pore ratios larger than 1. I would prefer to assign them to the physical limits (0 or 1).

---

## Author Comment (AC1) · 31 Mar 2020

We thank Christo Buizert for his comments on the manuscript and his suggestions for its improvement.

We have copied his comments below in blue, with our responses in black below.

Best Regards,
Kévin Fourteau on behalf of all co-authors.

Fourteau et al. publish detailed historical firn pycnometry records from three sites (Vostok, Greenland Summit and Law Dome DE-08) that were never published due to the passing of Jean-Marc Barnola. A porosity parameterization based on these data (published in Goujon et al. 2003) is widely used, and the paper describes the underlying data. These are very valuable observations that need to be published, and I commend Fourteau et al. for taking the effort to do so. I only have some recommendations for technical corrections.

I am very intrigued to see the differences in the close-off density at these three sites (Fig. 3). This suggests highest air content at DE08, and lowest at Vostok. This is indeed the observation of Martinerie et al. (1994). Do you think this is the underlying mechanism of the Martinerie et al. relationship between air content and temperature? This is certainly what is implied by Fig. 5. I think it would be appropriate to address this briefly in the paper.

Martinerie et al (1994) report that not only air content increases with temperature, but also that the porous volume at close-off (V_i in their article) increases with temperature. This is consistent with the pycnometry data presented here, with warmer sites closing at lower densities and therefore with a larger porous volume. This is however inconsistent with the results of Schaller et al (2017), essentially showing that the porous volume at close-off is site-independent. Further work is required to explain this discrepancy, and goes beyond the scope of this paper.

We will address this point in the paper **P5L30**:

*"It is interesting to note that the pycnometry data indicate that a cold site like Vostok reaches pore close-off at a higher density than a warm site like DE08-2. This is consistent with the results of Martinerie et al. (1994) that indicate an increase of porous volume at close-off with temperature based on air content measurements."*

Would it be possible to show the newer pycnometry data from the "Lock-In" site (Fourteau et al. 2019) somewhere together with the data from the three old sites (for example as a third panel in Fig. 3)? One would expect them to look more like Vostok than like Summit or DE08. Is that indeed the case? Showing these data would also give us a way to compare the consistency of the older and newer data.

[Figure]

As seen in the Figure of this response, the Lock-In pycnometry data are closer to the Summit values. However, one should be aware that the Lock-In data have not been submitted to an alpha correction similar to the datasets presented here, as this was not part of the experimental protocol during the Lock-In measurements in 2018. However, the Lock-In pycnometry data have been confirmed using tomography scanning of some of the samples.

On the other hand, if one uses air content data to derive the porous volume at close-off, the close-off at Lock-In is expected to happen at a porosity closer to the one of Vostok. This indicates that a discrepancy subsists between the close porosity data and the measured air content values.

These are for sure interesting questions that deserve dedicated work. However, we think that discussing them in this short article will obscure our main goal here, which is to made these historical data accessible, along with the protocol used to obtain them.

Could you give some examples of mechanisms that can cause system drift in the pynometry measurements? Are you talking about vapor freezing onto the surfaces, snow getting stuck, etc? Or is it something else?
We were not able to specifically identify the physical mechanism at the origin of this system drift, and J.M. Barnola notebooks did not provide any hypothesis for it neither. Our understanding is that this missing mechanism is one of the reasons why J.M. Barnola did not publish the data in the 90's.

Figure 1 of the article shows that the alpha coefficient tends to be lower than one. This indicates that the system drift usually leads to an overestimation of the inaccessible volume Vs. This overestimation could be due to an overestimation of the volume chamber V2. One could imagine that this overestimation of V2 comes from the presence of firn dust or frozen surfaces in the chamber, that would decrease its actual volume. However, this is largely speculative at this point as we do not have any supporting observations. We will write clearly in the text that we do not know the physical mechanism responsible for the system drift, **P4L30**:
*"One should note that we were not able to identify the physical mechanism at the origin of the pycnometry system drift. Further work should be dedicated to this topic."*

We will correct the technical points below, following the recommendations of the referee.

P1L8: remove "intend to"

P1L20: remove plural "s" from "precipitations"

P2L6 and P2L7: replace "reach" with "connect"

P2L6: airtight is not the right word, given that fugitive gases like Ne still escape. I think you can just remove the phrase "and are therefore airtight".

P4L4: "on the field" should be "in the field"

P5L2: "depending of" should be "depending on"

---

## Author Comment (AC2) · 31 Mar 2020

We thank Johannes Freitag for its constructive review, and for his suggestions to improve the clarity of the paper and the dataset.
We have copied his comments below in blue, with our responses in black.

Best Regards,
Kévin Fourteau on behalf of all coauthors.

Fourteau et al present historical porosity data of firn from three polar sites originally measured by J.-M. Barnola with the means of gas pycnometry several years ago. These data were the base of the important closed porosity-density parametrization widely used in firn gas models. Fourteau et al. provide the raw data and give insight into the original data processing. They confirm the use of the dataset for the closed-porosity parametrization first introduced in Goujon et al. (2003) and highlight its limitation. I appreciate the reworking of this fundamental data set.

The manuscript is well written with an appropriate introduction and description of the method. The data processing is described in required detail to enable the reader to follow the analysis. The data set is accessible via the given identifier and complete. I highly recommend the manuscript for publication. In my opinion the data set will encourage further methodological improvements and investigations on the cut-bubble-effect. Maybe it would be worthwhile to set up a future study where different methods like X-ray tomography and pycnometry are applied on the same set of samples?
We strongly agree that a joint study of pycnometry and large-scale tomography is currently one of the best option to study the cut-bubble effect in firn samples, and to check the consistency between pycnometry and tomography-based data.

Specific comments: 1) The names of the columns in the data set are a little bit misleading. The pore volumes are named by "Poros_xx" like "Poros_closed_cm__3_" which might be interpreted as abbreviation for porosity (no units) instead of pore volumes (unit in cm_3). I would suggest names like "Pore_vol_xx". 2) The column "Pores_frac" should be renamed to "ClosedPorosRatio" as it is defined in the manuscript. 3) In the data sets there are some non-physical values like negative pore volumina or closed pore ratios larger than 1. I would prefer to assign them to the physical limits (0 or 1).
Unfortunately, it is ESSD policy to have the data registered before the submission of the paper and PANGAEA policy not to modify datasets once they are registered. We will therefore not be able to modify the data submitted to the PANGAEA database. We will add a sentence in the article highlighting the difference in naming between the database and the article **P9L10**:
*"Finally, we made these data publicly available on the PANGAEA database (Fourteau et al., 2019a). Note that the naming convention used in the database is different from the one used in the article."*

We however do not agree that the data should be clipped to only have closed porosity ratios between 0 and 1. This would artificially reduce the experimental dispersion of the pycnometry method and introduce a bias in the data for fully open samples (respectively fully closed), as positive errors would no longer be statistically compensated by negative errors (respectively negative errors compensated by positive errors). We will add a sentence clarifying the presence of non-physical values in the dataset **P7L3**:
*"Finally, because of experimental dispersion, some firn samples were measured with a closed pore volume below zero or above the total porous volume. Potential users of the data should be aware that these values are not physically sound, and reflects the experimental errors of the pycnometry method."*

---

## Author Response (AR2)

Dear Pr Peng,

Please find a version with the required correction highlighted in blue (P9L2).
Please also note that we have completed the acknowledgement section, as we forgot to do when we submitted the previous revised version of the manuscript.

Best Regards,
Kévin Fourteau, On behalf of all co-authors

[revised manuscript text omitted]